# Optimization of Hydrogen Peroxide Concentrations for Inducing Oxidative Stress in Bovine Oocytes Prior to *In Vitro* Maturation

**DOI:** 10.3390/ani15223304

**Published:** 2025-11-16

**Authors:** Sirawit Yindeetrakul, Supawit Triwutanon, Anawat Sangmalee, Theera Rukkwamsuk

**Affiliations:** Department of Large Animal and Wildlife Clinical Sciences, Faculty of Veterinary Medicine, Kasetsart University, Kamphaeng Saen Campus, Kamphaeng Saen, Nakhon Pathom 73140, Thailand; sirawit.yin@ku.th (S.Y.); supawit.tr@ku.ac.th (S.T.); anawat.s@ku.th (A.S.)

**Keywords:** bovine oocyte, *in vitro* maturation, oxidative stress, reactive oxygen species

## Abstract

Oxidative stress impairs the quality of oocytes and limits the success of *in vitro* embryo production in cattle. A reliable model that mimics oxidative damage under controlled laboratory conditions is developed. In this study, bovine oocytes were exposed to different concentrations of hydrogen peroxide (H_2_O_2_) to evaluate oxidative stress without irreversible cellular damage. The findings revealed that short-term exposure to a specific concentration of H_2_O_2_ could consistently disrupt oocyte maturation, providing a simple and reproducible way to simulate oxidative stress *in vitro*. Importantly, this pre-*in vitro* maturation exposure model also reflects the physiological stress that oocytes may commonly experience. This model can serve as a valuable tool for future research, aiming at improving oocyte resilience and optimizing antioxidant treatments in animal reproductive biotechnology.

## 1. Introduction

The use of assisted reproductive technologies to increase the population and enhance the genetic quality of Thai beef cattle is essential for improving their economic value and preserving indigenous breeds [1]. Among these technologies, *in vitro* embryo production (IVEP) stands out as particularly effective. The IVEP process involves several key steps, starting with oocyte collection either from ovaries obtained at slaughterhouses or through ovum pick-up (OPU) from live animals. The collected oocytes undergo *in vitro* maturation (IVM), followed by *in vitro* fertilization (IVF) and *in vitro* culture (IVC) until they develop to the morula or blastocyst stage. These embryos can then be transferred to synchronized recipients or cryopreserved for future use [2]. Several factors influence oocyte quality, including donor characteristics, nutritional status, and environmental conditions, which collectively affect follicular development and oocyte competence [3]. Another critical factor is oxidative stress, which affects oocyte quality and the progression of oocyte maturation and pre-implantation embryo development. This stress can occur at any stage from the pre-culture phase through the entire IVC process [4].

Oxidative stress in oocytes can arise from both endogenous and exogenous sources [5]. Endogenous factors are primarily linked to the physiological status of the donor animal. After slaughter, ischemia and oxygen deprivation in ovarian tissues can impair mitochondrial function [6], leading to excessive production of reactive oxygen species (ROS) such as superoxide anion (O_2_^●−^), hydrogen peroxide (H_2_O_2_), and hydroxyl radicals (HO^●^). This imbalance triggers lipid, protein, and DNA oxidation [4] and disrupts microtubule polymerization within the spindle apparatus, thereby hindering proper meiotic progression and maturation [7]. The ROS production can persist throughout all stages of IVEP, starting approximately 4 h after IVM and peaking after 12 h of culture [8]. This increase coincides with the oocyte’s heightened ATP demand during critical processes, such as spindle formation, chromosome segregation, and cell division [9]. Exogenous factors include environmental and handling conditions during *in vitro* procedures. Elevated oxygen tension compared to *in vivo* conditions, temperature fluctuations, light exposure during oocyte handling, and trace transition metals (Fe^2+^, Cu^2+^) in culture media can enhance ROS generation via Fenton-type reactions [10].

Given the detrimental effects of oxidative stress on oocyte quality and subsequent embryo development, establishing a reliable *in vitro* oxidative stress model is essential. Among the various agents used to induce oxidative stress, hydrogen peroxide (H_2_O_2_) is one of the most employed agents due to its stability, ability to permeate cell membranes, and capacity to generate secondary reactive species such as hydroxyl radicals via Fenton reactions [11].

However, the concentration of H_2_O_2_ used is a critical factor. While physiological levels of ROS are known to play a regulatory role in meiotic resumption and oocyte maturation, excessive ROS can impair chromosomal alignment, damage the spindle apparatus, promote apoptosis, and ultimately compromise oocyte quality and embryo development [12]. For example, Zhang et al. [7] reported that 50–100 µM H_2_O_2_ effectively induced mitochondrial dysfunction and spindle disorganization in mouse oocytes in a dose- and time-dependent manner. Therefore, it is essential to select an appropriate H_2_O_2_ concentration that is high enough to induce a measurable cellular response without causing irreversible damage or cell death, ensuring that the model reflects physiologically relevant stress conditions. In addition, the 1 h-period of exposure was chosen as a fixed time point based on evidence that oxidative injury induced by H_2_O_2_ is both dose- and time-dependent [13]. This fixed duration allowed the assessment of concentration-dependent effects, assuming that at a given time, an optimal H_2_O_2_ concentration can induce measurable oxidative stress without causing any irreversible damage. The establishment of such an optimized oxidative stress model provides a valuable experimental framework for future studies on evaluating the efficacy of antioxidant compounds. By identifying a H_2_O_2_ concentration that induces a controlled reduction in nuclear maturation—specifically, a measurable decline in the proportion of oocytes reaching the MII stage—this model can reliably simulate oxidative challenge under *in vitro* conditions. Moreover, the use of a pre-IVM exposure design is particularly relevant because it mimics the transient oxidative stress that oocytes may naturally encounter during collection, transportation, or handling prior to culture. This approach allows for the consistent and reproducible assessment of protective agents, thereby enhancing the accuracy and interpretability of antioxidant efficacy studies in bovine oocytes. Based on this rationale, the present study aimed to determine the optimal H_2_O_2_ concentration for inducing oxidative stress in bovine oocytes during the pre-IVM phase.

## 2. Materials and Methods

### 2.1. Ovaries Collection and Handling

The oocytes used in this study were collected from ovaries of intact mature female cattle slaughtered at three slaughterhouses in Nakhon Pathom province. Oocyte collection was conducted between June and September 2025. The bovine ovaries were transported to the laboratory within 2 h of collection. During transportation, the ovaries were maintained at 30–35 °C in 0.9% NaCl supplemented with 100 IU/mL penicillin and 100 µg/mL streptomycin sulfate (HyClone™, Logan, UT, USA) to minimize microbial contamination.

### 2.2. Cumulus-Oocyte Complexes Recovery and Selection

Upon arrival at the laboratory, ovaries were washed three times with pre-warmed (37 °C) transport medium. Cumulus-oocyte complexes (COCs) were then aspirated from follicles measuring 2–5 mm in diameter using an 18-gauge needle attached to a 10 mL disposable syringe. Only COCs of grade 1 and 2 quality were used, characterized in accordance with the International Embryo Technology Society (IETS)

### 2.3. Induction of Oxidative Stress and In Vitro Maturation

Prior to IVM, COCs were randomly allocated (block-randomized by ovary) into four experimental groups based on the intended H_2_O_2_ concentration: 0 µM (control), 50 µM, 75 µM, and 100 µM. Each treatment group was conducted in five independent replicates, using oocytes collected from different ovaries for each replicate. Approximately 20 to 22 COCs were allocated to each treatment group per replicate. All groups were transferred using a mouth pipette into a collecting medium composed of phosphate-buffered saline (PBS) supplemented with 10% fetal bovine serum (FBS; HyClone™, Cytiva, Marlborough, MA, USA), 100 IU/mL penicillin, and 100 µg/mL streptomycin (HyClone™, Cytiva, USA). For the H_2_O_2_-treated groups (50, 75 and 100 µM), hydrogen peroxide (H_2_O_2_; 30% *w*/*w* analytical grade, ACS reagent, Fisher Scientific, Waltham, MA, USA) was freshly added to the collecting medium to achieve the target final concentrations. The COCs were then incubated at 38.5 °C in a humidified atmosphere of 5% CO_2_ for 1 h to induce oxidative stress. The control group was handled identically except without the addition of H_2_O_2_. Following the treatment period, COCs from all groups were washed three times in maturation medium to remove residual H_2_O_2_, and then selected COCs were placed into 50 µL drops of maturation medium at a ratio of 10 oocytes per drop. The oocytes were then cultured at 38.5 °C under a humidified atmosphere of 5% CO_2_ in air for 23 h. The maturation medium was based on Medium-199 (M-199; Cytiva, USA) supplemented with 10% FBS, 10 IU/mL follicle-stimulating hormone (Folligon; Intervet, Boxmeer, The Netherlands) 10 IU/mL human chorionic gonadotropin (Chorulon; Intervet, The Netherlands), 100 IU/mL penicillin, and 100 µg/mL streptomycin sulfate (HyClone™, Cytiva, USA). All media were pre-equilibrated for 2 h in a humidified CO_2_ incubator prior to use.

### 2.4. Aceto-Orcein Staining

Following incubation, cumulus cells were removed by repeated pipetting in PBS containing 0.25% hyaluronidase (Medchem Express, Monmouth Junction, NJ, USA). The denuded oocytes were then fixed in methanol-acetic acid (1:3 *v*/*v*) for 48 h. After fixation, oocytes were stained with 2% aceto-orcein (prepared in 45% acetic acid) for 10 min. The nuclear maturation stage of each oocyte was subsequently assessed under a phase-contrast microscope (Olympus CX22LED, Olympus, Tokyo, Japan) at 400× total magnification.

### 2.5. Data Analysis

The normality of data distribution was assessed using both Shapiro-Wilk and Kolmogorov- Smirnov tests. All variables satisfied the assumption of normality. Homogeneity of variance was verified using Levene’s test before conducting further analysis. Comparisons between the means among the four studied groups were conducted using one-way ANOVA. Multiple comparisons between each group were evaluated using Tukey’s Honestly Significant Difference (HSD) test.

## 3. Results

To evaluate the effect of oxidative stress on the nuclear maturation competence of bovine oocytes, COCs were cultured for 23 h under the standard IVM condition, after which the oocytes were assessed for nuclear status. Following fixation and staining with aceto-orcein, chromatin configuration was examined under a phase-contrast microscope. The nuclear maturation stage of each oocyte was classified as shown in Figure 1.

Exposure of bovine oocytes to H_2_O_2_ during the pre-IVM phase resulted in a dose-dependent trend of reduced meiotic maturation, which became statistically significant only at the highest concentration tested. The control group (0 µM) exhibited the highest proportion of oocytes reaching the MII stage (69.23 ± 8.45%), which was not significantly different from the 50 µM (67.5 ± 12.29%) or 75 µM (56.5 ± 2.33%) groups (*p* > 0.05). In contrast, exposure to 100 µM H_2_O_2_ significantly reduced the proportion of MII-stage oocytes (32.83 ± 7.64%) compared with all other groups (*p* < 0.05). These findings suggested that mild oxidative stress (75 µM H_2_O_2_) did not significantly impair nuclear maturation, whereas higher oxidative stress (100 µM) severely compromises meiotic progression. The detailed distributions of oocytes across nuclear maturation stages (GV, GVBD, MI, and MII) for each treatment group are presented in Table 1, and the corresponding dose–response relationship between H_2_O_2_ concentration and MII rate is illustrated in Figure 2.

## 4. Discussion

In this study, oxidative stress induced by H_2_O_2_ prior to IVM has been demonstrated to have a substantial negative impact on oocyte quality, significantly reducing the proportion of oocytes that progressed to the MII stage.

The underlying mechanism by which oxidative stress impairs oocyte maturation is believed to involve disruption of multiple cellular processes, when ROS levels exceed physiological thresholds. Excessive ROS can destabilize the maturation-promoting factor (MPF), induce premature degradation of cyclin B1, and increase cytosolic calcium levels. These alterations activate the calcium/calmodulin-dependent protein kinase II-anaphase-promoting complex/cyclosome (CaMKII-APCC/C) pathway, potentially resulting in abnormal meiotic exit or arrest at the MI stage, thereby preventing normal meiotic progression [14]. In addition, high ROS levels can directly damage key cellular structures by inducing lipid peroxidation of mitochondrial membranes [15] and causing defects of the spindle apparatus and chromosomal misalignment [7]. Such damage also activates cell death pathways, including apoptosis and autophagy, via mechanisms involving the ratio of B-cell lymphoma-2-associated X protein (Bax) to B-cell lymphoma 2 (Bcl-2) signaling, cytochrome c release, and caspase cascade activation [16]. Collectively, these alterations significantly compromise the fertilization competence of the oocyte and its ability to support normal embryo development [17]. In this study, the exposure of bovine oocytes to increasing H_2_O_2_ concentrations revealed a dose-dependent trend in meiotic inhibition. The absence of a significant decrease in MII rate at 75 µM H_2_O_2_ may indicate that this concentration induced only mild oxidative stress, activating endogenous antioxidant defenses (e.g., superoxide dismutase, catalase, and glutathione peroxidase) without surpassing the oocyte’s compensatory threshold. This partial activation could maintain sufficient MPF stability and mitochondrial function to allow nuclear maturation. In contrast, exposure to 100 µM H_2_O_2_ resulted in a marked accumulation of germinal vesicle (GV)-stage oocytes, suggesting that excessive ROS interfered with meiotic resumption by inhibiting GVBD initiation. This effect may arise from oxidative modification of CDK1 or other MPF subunits required for GVBD activation, consistent with previous findings that high ROS concentrations suppress cyclin B1 accumulation and block nuclear envelope breakdown [18].

While the cytotoxic effects of high ROS concentrations are well documented, some studies have also highlighted the physiological role of ROS at low-to-moderate levels. For example, short-term exposure of mature bovine oocytes to 50–100 µM H_2_O_2_ for 1 h has been shown to enhance blastocyst yield and reduce apoptosis in the resulting embryos [19]; however, the same treatment conditions in this study produced inhibitory effects. This discrepancy likely stems from differences in experimental design, including the developmental stage at exposure and the cellular redox state at the time of treatment. Similarly, low concentrations of H_2_O_2_ have been reported to induce GVBD in immature oocytes. However, higher concentrations inhibit first polar body extrusion and trigger apoptosis, characterized by elevated Bax expression, DNA fragmentation, and caspase-3 activation [13]. These findings support the dual role of ROS, not only as harmful byproducts of metabolism but also as critical signaling molecules that regulate key events in oocyte maturation. At physiological levels, ROS may contribute to the activation of maturation-promoting factor, facilitating GVBD and progression through meiosis stages to metaphase I and II [12,17].

Oxidative stress is known to disrupt mitochondrial function, leading to a decline in ATP production that compromises microtubule polymerization and spindle organization during meiosis. Such mitochondrial dysfunction results in spindle disassembly, chromosome misalignment, and impaired cytoskeletal stability, ultimately reducing oocyte developmental competence [7]. Consistent with this, oocytes exposed to systemic oxidative stress or metabolic disorders exhibit enlarged and disorganized spindles, misaligned chromosomes, and reduced maturation and blastocyst formation rates, reflecting a direct relationship between cytoskeletal abnormalities and developmental potentials [20]. Proper meiotic division relies on the coordination between mitochondrial activity and microtubule organization, which ensures accurate spindle assembly and chromosome segregation [21]. Abnormal spindle morphology, resulting from impaired cytoskeletal regulation, has been associated with reduced fertilization potential and a lower rate of euploid blastocyst formation [22]. Although the present study successfully characterized the oxidative threshold that compromises oocyte maturation, it did not assess intracellular oxidative status or meiotic spindle morphology. Future investigations integrating ROS quantification and spindle imaging are therefore warranted to determine whether oxidative stress at sub-lethal concentrations may still induce subtle cytoskeletal alterations that impair subsequent fertilization or embryo development.

## 5. Conclusions

This study developed a reproducible *in vitro* model capable of inducing oxidative stress in bovine oocytes through exposure to hydrogen peroxide (H_2_O_2_) prior to the IVM phase. The results demonstrated that treatment with 100 µM H_2_O_2_ for 1 h effectively induced a measurable oxidative imbalance without causing irreversible cellular damage, making it an appropriate model for evaluating the efficacy of antioxidant supplementation under controlled stress conditions. Beyond serving as an experimental oxidative challenge, this pre-IVM H_2_O_2_ exposure model also simulates the oxidative stress that oocytes may experience during pre-culture handling such as follicular aspiration, transportation, or laboratory processing, thereby increasing its physiological relevance. This model provides an essential foundation for future investigations aimed at assessing the functional roles of antioxidants in protecting oocytes from oxidative damage. Future research should focus on further refinement of this model by integrating analyses of intracellular redox status, mitochondrial activity, and spindle morphology to achieve a more comprehensive understanding of how oxidative imbalance affects oocyte physiology. Such knowledge will contribute to the development of improved antioxidant supplementation strategies and optimized culture conditions to enhance oocyte competence and increase the efficiency of embryo production in assisted reproductive systems.

## Figures and Tables

**Figure 1 animals-15-03304-f001:**
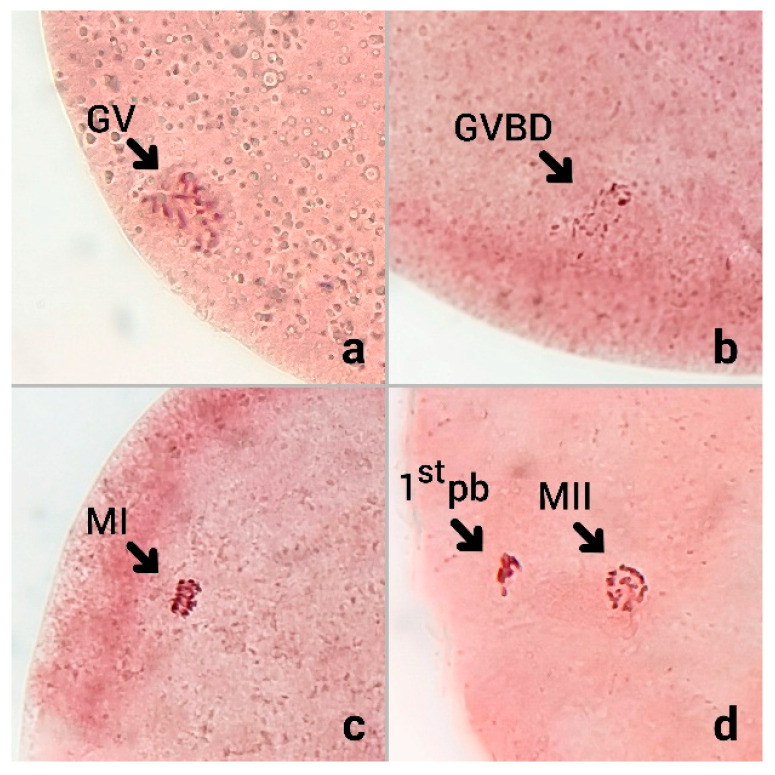
The nuclear status of bovine oocytes after 23 h of *in vitro* maturation (IVM), stained with aceto-orcein. (**a**) Germinal vesicle (GV): intact nuclear membrane with visible germinal vesicle. (**b**) Germinal vesicle breakdown (GVBD): nuclear envelope dissolution with condensed chromatin. (**c**) Metaphase I (MI): chromosomes aligned on the first meiotic spindle. (**d**) Metaphase II (MII): mature oocyte with first polar body (1st pb) extruded, indicating successful completion of meiosis I and arrest at MII, assessed under a phase-contrast microscope at 400× total magnification.

**Figure 2 animals-15-03304-f002:**
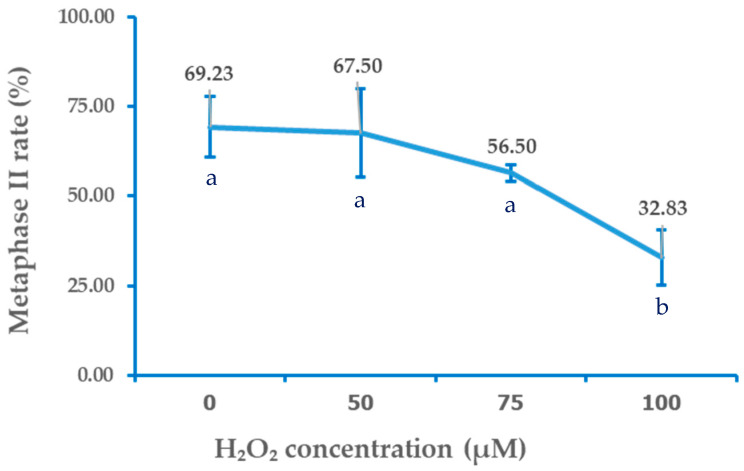
Dose-response curve of H_2_O_2_ concentration and metaphase II MII) rate. Data are mean and the error bars are SD. Different letters indicate the means at each concentration significantly differ at *p* < 0.05.

**Table 1 animals-15-03304-t001:** Percentage of bovine oocytes at different meiotic stages (GV, GVBD, MI, MII) after 23 h of *in vitro* maturation under three treatment groups. Results show the mean ± SD of percentage with different superscripts in the same column differ significantly (*p* < 0.05). GV= germinal vesicle, GVBD = germinal vesicle breakdown, MI = metaphase I, MII = metaphase II. Numbers of oocytes examined were obtained from five independent replicates.

H_2_O_2_ Exposed	Number of Oocytes Examined	Percentage of Oocytes at Each Stage			
		GV	GVBD	MI	MII
0 µM	101	0 ^a^	14.47 ± 7.50 ^a^	16.30 ± 10.06 ^a^	69.23 ± 8.45 ^a^
50 µM	105	1.85 ± 3.70 ^a^	17.41 ± 16.87 ^a^	13.24 ± 7.52 ^a^	67.50 ± 12.29 ^a^
75 µM	104	6.19 ± 20 ^a^	11.86 ± 5.48 ^a^	25.44 ± 4.33 ^a^	56.50 ± 2.33 ^a^
100 µM	107	20.64 ± 16.70 ^b^	25.29 ± 10.76 ^a^	21.22 ± 10.00 ^a^	32.83 ± 7.64 ^b^

## Data Availability

The data presented in this study are available from the corresponding author upon reasonable request. No publicly archived datasets were generated or analyzed during the current study.

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
