# Peer review of "Optimization of Hydrogen Peroxide Concentrations for Inducing Oxidative Stress in Bovine Oocytes Prior to In Vitro Maturation"

_animals, 2025, doi:10.3390/ani15223304_

Round 1

Reviewer 1 Report (Previous Reviewer 1)

Comments and Suggestions for Authors

Thank you for taking under consideration my comments. The quality of the manuscript has been significantly improved.

Suggestions

Please explain abbreviation such as Bax, Bcl etc. and use a table with the abbreviations used in the text. 

In Figure 2, an asterisk at 100μΜ would be helpful for the reader. 

The explanation regarding the choise of time and concentrations should be added in the text in the introduction section , and not just as an answer to the Reviewer's comments. 

Author Response

Reviewer 2 Report (Previous Reviewer 2)

Comments and Suggestions for Authors

The authors have made comprehensive revisions according to the reviewers' comments. It is suggested to accept and publish the manuscript after minor revisions.

1) Significant letter markers (e.g., a, b) need to be added to the data of each group in Figure 2 to be consistent with Table 1, so as to intuitively present the differences between groups.

2) "uM" and "µM" are mixed in the text and need to be unified as "µM"; The subscript and superscript format of "Hâ‚‚Oâ‚‚" must be consistent throughout the text (e.g., avoid the mixed use of "Hâ‚‚Oâ‚‚" and incorrect formatting versions of "Hâ‚‚Oâ‚‚").

Author Response

This manuscript is a resubmission of an earlier submission. The following is a list of the peer review reports and author responses from that submission.

Round 1

Reviewer 1 Report

Comments and Suggestions for Authors

Dear authors,

Thank you for taking the effort and the time to submit your article “Optimization of Hydrogen Peroxide Concentrations for Inducing Oxidative Stress in Bovine Oocytes Prior to In vitro Maturation”.

The article has been prepared according to the format and guidelines of the journal. The photographs are well-prepared and informative and effectively demonstrate the experimental findings. The language is clear, allowing the scientific background and results to be easily understood. However, according to my scientific opinion there is a gap in the interpretation of the hypothesis and the limitations of the study:

  1. An important limitation of the study is the absence of data on the oxidative status of the oocytes and on meiotic spindle morphology. Both oxidative balance and spindle integrity are key determinants of cytoplasmic and nuclear maturation, directly influencing fertilization potential and early embryonic development. In addition, the study did not evaluate the association between spindle abnormalities and in vitro fertilization success, which could help explain whether lower concentrations of the tested agent, despite supporting nuclear maturation, might still impair fertilization through subtle cytoskeletal disturbances. This limitation should be discussed. Here, you can find some relative references to support my recommendation:
  • Hinrichs et al., 1993; Lonergan et al., 2003; Combelles et al., 2011:
    Bovine oocytes with normal spindles and aligned chromosomes had significantly higher fertilization and blastocyst rates than those with spindle defects.
  • Lorenzo et al., 2020: Oocytes showing spindle disorganization post-IVM had reduced pronuclear formation and lower blastocyst quality even if fertilization occurred.
  • Fair et al., 2005: The presence of spindle and chromosomal abnormalities correlated with lower developmental competence and increased embryonic arrest before the 8-cell stage.
  1. The authors did not explain why is it important to establish a model for the induction of oxidative stress. The hypothesis is unclear since the results are not correlated with other biological markers, eg the antioxidant status of oocytes, spindle abnormalities, apoptosis markers etc.
  2. How did you choose these concentrations? And why did you only incubate oocytes for 1h? Oxidative stress is a dynamic procedure, since ROS accumulate in the media due to cellular metabolism.

Below, there is my point to point evaluation of your manuscript:

  1. The simple summary looks like a revised version of the abstract.
  2. Many sentences need reference, eg. L.41-46, L. 55-60, L. 185-196 etc.
  3. L. 41 The use of assisted reproductive technologies
  4. L. 42 Does the economic value imply the genetic improvement of the breed?
  5. L. 51 Does the effect of photoperiod refer to bovine species? Clarify
  6. L. 52 change , to .
  7. L. 52 You should categorize the factors that contribute to oxidative stress to endogenous and exogenous factors.
  8. L. 62 ROS production can persist throughout all the stages of IVEP. Please revise accordingly.
  9. L. 70 You may consider revising this part for clarity since it is quite vague. The last factor (ischemia) could be presented first, as it represents a general condition related to the animal’s physiological status rather than to the IVF procedure itself. Additionally, you could elaborate on how this factor may affect oocyte quality through mechanisms involving oxidative stress, which has been shown to impair meiotic progression and cytoplasmic maturation (see for example, Agarwal et al., 2012, Reproductive Biology and Endocrinology, 10:49).
  10. L. 102 Add a reference for the criteria.
  11. Section 2.4. should be moved in Line 118.
  12. L. 142 Results??

Reviewer 2 Report

Comments and Suggestions for Authors

This study focuses on the construction of an oxidative stress model for bovine oocytes prior to in vitro maturation (IVM), using hydrogen peroxide (H2O2) as the oxidative stress inducer and screening for optimal induction conditions through gradient concentration experiments. The research topic aligns with the need for studies on oxidative stress regulation in the field of bovine in vitro embryo production (IVEP), and holds clear theoretical value and application prospects. The study design follows a logical framework, with a complete methodological structure, and the data results can support the core conclusions. However, there is room for optimization in aspects such as the completeness of experimental details, the accuracy of data presentation, the depth of mechanism discussion, and the standardization of content expression.

Specific Review Comments:

  1. Supplementary Information on Sample Size and Number of Replicates

The current methodology only mentions that "the number of oocytes in each group was balanced across biological replicates" but fails to specify the number of biological replicates (e.g., number of independent experimental rounds) and the specific number of oocytes allocated to each group in each experimental round. Sample size and number of replicates are critical for ensuring the reliability of experimental results; thus, the following supplements are recommended:â‘ The specific number of biological replicates (e.g., 3 or 5 independent experiments); â‘¡ The initial number of oocytes in each group (0 μM, 50 μM, 75 μM, 100 μM) per experimental round, as well as the effective number of oocytes used for staining analysis (it is necessary to explain whether abnormal oocytes were excluded and the exclusion criteria).

  1. Missing Information on Key Reagents and Instruments

Specifications and parameters of some key reagents in the experiment are not clearly stated, which may affect the reproducibility of the experiment:â‘  Purity grade (e.g., analytical grade, guaranteed reagent grade) and manufacturer of hydrogen peroxide (\(H_2O_2\));â‘¡ Batch number and source of fetal bovine serum (FBS) (compositional differences between FBS batches may affect oocyte status);â‘¢ Model and imaging resolution of the phase-contrast microscope.It is recommended to supplement the above information in the corresponding sections.

  1. Unclear Oocyte Grading Standards

The methodology mentions that "only grade 1 and grade 2 oocytes were selected" but does not define the specific criteria for classifying grade 1 and grade 2 oocytes (e.g., quantitative description of cytoplasm uniformity, exact range of cumulus cell layers, presence of granulosa cell debris, etc.). Oocyte grading may vary across laboratories; clarifying the standards can avoid biases caused by subjective judgment. It is recommended to refer to industry universal standards (e.g., standards of the International Embryo Technology Society, IETS) to supplement detailed grading criteria.

  1. Vague Description of Statistical MethodsIn

"2.6 Data analysis", only "one-way analysis of variance (one-way ANOVA) was used to compare means between groups, and Post Hoc tests were used for multiple comparisons" is mentioned, without specifying:â‘  The specific type of Post Hoc test (e.g., Tukey’s HSD, Duncan’s multiple range test, etc., as different test methods have different significance thresholds);â‘¡ The specific method for testing data normality (e.g., Shapiro-Wilk test, Kolmogorov-Smirnov test);â‘¢ Results of homogeneity of variance test (if variance is heterogeneous, it is necessary to explain whether a correction method such as Welch’s ANOVA was adopted).It is recommended to supplement the above statistical details to ensure the rigor of the statistical methods.

  1. Errors and Format Issues in Table Data

Table 1 (Percentage of bovine oocytes at different meiotic stages) contains obvious expression errors:â‘  In the header "Percentage of oocyte at each stage", "oocyte" should be in the plural form "oocytes";â‘¡ The headers of the data columns "GV", "GVBD", "MI", and "MII" are misaligned with the content (in the current table structure, columns "GV", "GVBD", etc., directly follow the "Hâ‚‚Oâ‚‚ exposed" column, with no clear correspondence between column titles and content. The table format needs to be adjusted to ensure that each column of data accurately corresponds to the stage name);â‘¢ Some data units or symbols are incorrect (e.g., "20.64±16.7", "25.29±10.76" in the 100 µM group). It is recommended to uniformly retain two decimal places to maintain consistency in data format.

  1. Inconsistency Between Result Description and Data

In "3.1 Oocyte maturation assessment", it is mentioned that "the MII rate of the 75 μM group (56.5±2.33%) decreased compared with the control group but without statistical significance", but there is no explanation of whether there is a statistical difference in the MII rate between this group and the 50 μM group (67.5±12.29%). Although the core conclusion is that the 100 μM group showed a significant decrease, supplementing the results of pairwise comparisons between all groups can more comprehensively reflect the dose effect.It is recommended to supplement the significance results of multiple comparisons between all groups in the results (e.g., 50 μM vs 75 μM: p=0.XX; 75 μM vs 100 μM: p=0.XX) and mark them with letters (e.g., a, b, c) in Table 1 to clarify the differences between groups.

  1. Lack of Visual Result Support

The results mention that "Figure 1 shows microscopic images of oocytes at different maturation stages" but do not provide specific content of Figure 1 (e.g., magnification of images at each stage, labeling of key structures such as polar bodies and chromosome arrangement). In addition, it is recommended to supplement a "dose-response curve of Hâ‚‚Oâ‚‚ concentration and MII rate" to intuitively show the decreasing trend of MII rate with increasing Hâ‚‚Oâ‚‚ concentration, thereby enhancing the readability of the results.

  1. Insufficient Depth of Mechanism Discussion

Although the discussion section mentions that "oxidative stress affects oocyte maturation by disrupting MPF, mitochondrial membrane potential, and activating apoptotic pathways", it fails to conduct targeted analysis combined with the data of this study:â‘  It does not explain why 75 μM Hâ‚‚Oâ‚‚ did not cause a significant decrease in MII rate (whether ROS at this concentration only slightly activates the antioxidant system without exceeding the cellular compensation threshold);â‘¡ It does not discuss the mechanism underlying the "increased proportion of GV-stage oocytes (20.64±16.7%) in the 100 μM Hâ‚‚Oâ‚‚ group" (whether ROS inhibits the initiation of meiosis, leading to GV-stage arrest).It is recommended to supplement targeted discussions on the above mechanisms by combining the stage distribution data of this study (changes in the proportions of GV, GVBD, and MI in Table 1) to enhance the relevance between the discussion and the results.

  1. Unmentioned Limitations of Conclusions

The conclusions only emphasize that "treatment with 100 μM Hâ‚‚Oâ‚‚ for 1 hour can be used as an oxidative stress model for bovine oocytes" but fail to point out the limitations of the model:â‘  The model only evaluates the oxidative stress effect based on nuclear maturation (MII rate) and does not detect cytoplasmic maturation indicators (e.g., mitochondrial distribution, glutathione (GSH) levels, direct detection of ROS content), making it impossible to rule out the possibility of "asynchrony between nuclear and cytoplasmic maturation";â‘¡ The model does not verify subsequent embryo developmental competence (e.g., cleavage rate and blastocyst rate after IVF), so it cannot confirm whether this stress condition affects embryonic developmental potential. It is recommended to supplement the above limitations in the conclusions to provide hints for future research directions (e.g., optimizing the model by combining cytoplasmic maturation indicators, verifying embryonic development effects).

  1. Disconnection Between Literature Citation and Discussion Logic

In the discussion, reference [9] (Zhou et al., 2016) mentions that "100 μM Hâ‚‚Oâ‚‚ affects goat oocyte maturation", but there is no comparison of differences between this study (bovine oocytes) and the goat study (e.g., differences in Hâ‚‚Oâ‚‚ sensitivity caused by species specificity); reference [11] (Vandaele et al., 2010) mentions that "short-term Hâ‚‚Oâ‚‚ treatment promotes bovine oocyte embryonic development", but there is no explanation for why only an inhibitory effect was observed in the 100 μM Hâ‚‚Oâ‚‚ group in this study (whether it is related to differences in treatment time or oocyte maturation stage).It is recommended to supplement comparative analysis between references to clarify the species specificity of the results of this study and the dependence on experimental conditions, thereby enhancing the logic of the discussion.(IV) Format and Expression Standardization1. Inconsistent Use of Symbols and UnitsIn the full text, the unit of Hâ‚‚Oâ‚‚ concentration is mixed between "uM" and "µM" (e.g., "100 uM", "50 µM"). "uM" is an non-standard expression and should be uniformly revised to the international standard symbol "µM" (micromoles per liter); in addition, "MIll rate" and "MIl stage" are typos and should be corrected to "MI rate" (metaphase I rate) and "MII stage" (metaphase II stage).It is recommended to proofread the full text to correct symbol errors and ensure the standardization of units.

  1. Paragraph Structure and Redundant Expression

Some paragraphs contain repeated expressions. For example, in "1. Introduction", content related to "oxidative stress affecting oocyte quality" is mentioned in both the first and third paragraphs and can be merged and condensed; in "3. Results", the description of "oocyte maturation stage classification" overlaps with the explanation of Figure 1, so it is recommended to delete redundant content and optimize paragraph logic. In addition, the expression "Exposure to 100 uM Hâ‚‚Oâ‚‚ for 1 hour effectively reduced meiotic maturation in a dose-dependent manner" in the Abstract is inaccurate and should be revised to "Exposure to 100 µM Hâ‚‚Oâ‚‚ for 1 hour significantly reduced meiotic maturation, showing a dose-dependent trend" to avoid logical confusion between "dose-dependent" and "significantly" (the first two concentrations showed no significant difference, and only the 100 μM group showed a significant decrease; the overall trend is a "dose-dependent trend" rather than a strict "dose dependence").
